# A Novel Urinary Proteomics Classifier for Non-Invasive Evaluation of Interstitial Fibrosis and Tubular Atrophy in Chronic Kidney Disease

**DOI:** 10.3390/proteomes9030032

**Published:** 2021-07-13

**Authors:** Lorenzo Catanese, Justyna Siwy, Emmanouil Mavrogeorgis, Kerstin Amann, Harald Mischak, Joachim Beige, Harald Rupprecht

**Affiliations:** 1Department of Nephrology, Angiology and Rheumatology, Klinikum Bayreuth GmbH, 95447 Bayreuth, Germany; lorenzoriccardo.catanese@gmail.com (L.C.); harald.rupprecht@klinikum-bayreuth.de (H.R.); 2Kuratorium for Dialysis and Transplantation (KfH) Bayreuth, 95445 Bayreuth, Germany; 3Friedrich-Alexander-University Erlangen-Nürnberg, 91054 Erlangen, Germany; 4Mosaiques Diagnostics GmbH, 30659 Hannover, Germany; mavrogeorgis@mosaiques.de (E.M.); mischak@mosaiques.de (H.M.); 5Institute for Molecular Cardiovascular Research (IMCAR), RWTH Aachen University Hospital, 52074 Aachen, Germany; 6Department of Nephropathology, Institute of Pathology, University of Erlangen-Nürnberg, 91054 Erlangen, Germany; Kerstin.Amann@uk-erlangen.de; 7Department of Infectious Diseases/Tropical Medicine, Nephrology/KfH Renal Unit and Rheumatology, St. Georg Hospital Leipzig, 04129 Leipzig, Germany; joachim.beige@kfh.de; 8Kuratorium for Dialysis and Transplantation (KfH) Renal Unit, Hospital St. Georg, 04129 Leipzig, Germany; 9Department of Internal Medicine II, Martin-Luther-University Halle/Wittenberg, 06108 Halle/Saale, Germany

**Keywords:** fibrosis, urine, peptides, IFTA, biomarkers

## Abstract

Non-invasive urinary peptide biomarkers are able to detect and predict chronic kidney disease (CKD). Moreover, specific urinary peptides enable discrimination of different CKD etiologies and offer an interesting alternative to invasive kidney biopsy, which cannot always be performed. The aim of this study was to define a urinary peptide classifier using mass spectrometry technology to predict the degree of renal interstitial fibrosis and tubular atrophy (IFTA) in CKD patients. The urinary peptide profiles of 435 patients enrolled in this study were analyzed using capillary electrophoresis coupled with mass spectrometry (CE-MS). Urine samples were collected on the day of the diagnostic kidney biopsy. The proteomics data were divided into a training (*n* = 200) and a test (*n* = 235) cohort. The fibrosis group was defined as IFTA ≥ 15% and no fibrosis as IFTA < 10%. Statistical comparison of the mass spectrometry data enabled identification of 29 urinary peptides with differential occurrence in samples with and without fibrosis. Several collagen fragments and peptide fragments of fetuin-A and others were combined into a peptidomic classifier. The classifier separated fibrosis from non-fibrosis patients in an independent test set (*n* = 186) with area under the curve (AUC) of 0.84 (95% CI: 0.779 to 0.889). A significant correlation of IFTA and FPP_BH29 scores could be observed Rho = 0.5, *p* < 0.0001. We identified a peptidomic classifier for renal fibrosis containing 29 peptide fragments corresponding to 13 different proteins. Urinary proteomics analysis can serve as a non-invasive tool to evaluate the degree of renal fibrosis, in contrast to kidney biopsy, which allows repeated measurements during the disease course.

## 1. Introduction

Proteomics-based techniques have been successfully used for the detection of specific biomarkers, with the possibility to describe the health status of individuals. Many proteomic studies have been performed in the context of different kidney diseases, providing valid and robust protein and peptide biomarkers for the diagnosis and prognosis of chronic kidney disease (CKD). The most commonly applied technique in this context is capillary electrophoresis coupled with mass spectrometry (CE-MS) with more than 800 manuscripts published within the last 20 years [1]. The application of this technique to analyze urinary peptide profiles in CKD patients was described in a review article in more detail [2]. Recent studies have proposed urinary proteome analysis as non-invasive liquid biopsy, providing the possibility to replace invasive kidney biopsy if it is not available or contraindicated [3]. Discriminating patients with CKD from healthy individuals, predicting progression of CKD, and distinguishing between different CKD etiologies was shown to be feasible using urinary proteome analysis [4,5,6]. Moreover, in the work of Magalhães et al. [3], the association of urinary peptides with renal interstitial fibrosis was described. Although interstitial fibrosis and tubular atrophy (IFTA) undoubtedly has a high prognostic value in all subtypes of CKD, its evaluation in routine diagnostic kidney biopsy currently lacks standardization [7]. CKD is one of the major global health burdens with a worldwide prevalence of 9–13% and an emerging risk factor for global morbidity and mortality, especially in countries with a low Socio-Demographic Index (SDI) [8,9]. The Kidney Disease: Improving Global Outcomes (KDIGO) CKD Work Group has defined CKD as an abnormal kidney structure or function present for more than 3 months with health implications [10]. The classification of CKD consists of three elements: cause, glomerular filtration rate (GFR) category, and albuminuria category. Together with comorbidities, this classification has been shown to allow risk stratification and prediction of CKD prognosis [11]. While GFR and albuminuria are easy to determine and are routinely assessed when evaluating a CKD patient, determining the cause of CKD can be more challenging. A thorough assessment of medical history, routine, and extended serological laboratory testing of blood and urine and functional diagnostics (i.e., ultrasound or computed tomography) are tools for evaluating CKD etiologies. Despite these tests, to achieve a high level of certainty regarding CKD etiology, histological analysis of kidney tissue is pivotal. Therefore, kidney biopsy remains a gold standard in the diagnosis of CKD etiologies and allows more accurate prediction of prognosis, therapy, and clinical outcome. One key feature assessed in most biopsies is interstitial fibrosis and tubular atrophy (IFTA). IFTA is a common final pathway of extracellular matrix (ECM) accumulation, which is a central pathogenetical mechanism with high contribution to functional loss in many chronic kidney diseases. It involves a variety of different cell types and numerous molecular pathways that lead to accumulation of collagen and related molecules in the interstitium [12]. IFTA has been used as a prognostic marker for CKD and is a hallmark for disease progression to end stage kidney disease (ESKD) [13]. Because of its prognostic value, IFTA has been incorporated into several scoring systems for CKD [14,15]. Unfortunately, due to its invasive nature, kidney biopsy cannot be routinely used for diagnosis of CKD. Kidney biopsy has a relatively high cost, due to hospitalization. In most cases, the need for a highly trained nephrologist who can carry out the procedure and a list of contraindications (bleeding diathesis, presence of a solitary native kidney, uncontrolled severe hypertension, anticoagulant or antiplatelet drugs) limits the availability of a percutaneous kidney biopsy as a diagnostic tool.

It was our aim to provide a solution to the problem of obtaining vital histological data to allow optimum clinical diagnosis. We investigated urinary peptide profiles and correlated them with corresponding histopathological findings regarding IFTA, thus offering a new non-invasive tool to quantify IFTA and facilitate prediction of progression of CKD into ESKD. Such a tool might contribute to or even substitute for kidney biopsy diagnostics.

## 2. Materials and Methods

### 2.1. Patient Cohort

Patient samples were collected in the Department of Nephrology of the Hospital Bayreuth GmbH (Germany) from 2008 to 2020. Samples were collected on the day of the diagnostic kidney biopsy. Clinical indication for biopsy was given beforehand independently of the study. Written consent for anonymized data retrieval and storage was obtained at least 1 day prior to the biopsy. The local ethics committee of the Friedrich-Alexander-Universität Erlangen-Nürnberg provided approval for the nephrological biobank of the Klinikum Bayreuth (ethic approval code 264_20 B) and the urinary proteomics analysis (ethic approval code 221_20 B). On the day of the kidney biopsy, venous blood was drawn and immediately analyzed for creatinine concentration and estimation of eGFR using Chronic Kidney Disease Epidemiology Collaboration (EPI) equation [16]. Multiple urine samples were obtained for assessment of proteinuria and cryo-stored for capillary electrophoresis and mass spectrometry. Biopsies were carried out and probes were sent to the Department of Nephropathology of the Friedrich-Alexander University Erlangen/Nürnberg for histopathological analysis.

A primary, and if applicable, a secondary histological diagnosis were extracted from the written histological report. Degree of IFTA was determined visually after histopathological staining through the Department of Nephropathology of the University of Erlangen-Nürnberg and given as percentage fibrotic vs. total interstitial area. The primary diagnosis of the biopsy was used for definition of patient groups. Transplant biopsies were excluded from further analysis.

The whole patient cohort consisting of 435 probe sets was divided in a low fibrosis group (*n* = 140; % IFTA < 10%) and a high fibrosis group 1 (*n* = 246; % IFTA ≥ 15%). Patients with IFTA percentages between 10 and 15% were excluded from primary analysis. The cut-off was chosen to also include early stages of fibrosis in the fibrosis group and exclude only patients with no relevant fibrosis. The cut-off was also chosen because of a rapid downfall of eGFR observed in our cohort around 10–20% IFTA, which indicates significant loss of kidney function possibly linked to fibrosis.

### 2.2. CE-MS Analysis

Technical details of the CE-MS analysis, including details on sample preparation, performance characteristics, reproducibility, etc. were described in detail in [17]. The technology was chosen due to its proven performance in routine applications in multiple studies, and also as a result of the availability of a large database for comparison [1] and since it was applied in large prospective clinical trials in the context of CKD [18]. For the CE-MS analysis, urine samples were thawed and 0.7 mL of urine were diluted with 0.7 mL of a solution containing 2M urea (VWR Chemicals, Leuven, Belgium), 10 mM NH_4_OH (Merc KGaA, Darmstadt, Germany), and 0.02% SDS (Carl Roth GmbH, Karlsruhe, Germany). The samples were ultrafiltered using a Centrisart ultracentrifugation filter device (20 kDa molecular weight cut-off; Sartorius, Goettingen, Germany). Subsequently, 1.1 mL filtrate was obtained and applied onto a PD-10 desalting column (GE Healthcare Bio Sciences, Uppsala, Sweden) equilibrated in 0.01% aqueous NH_4_OH. Finally, the eluate was lyophilized and stored at 4 °C prior to resuspension in HPLC-grade water for CE-MS analysis. CE-MS analysis of each individual sample was performed using a P/ACE MDQ capillary electrophoresis system (Beckman Coulter, Fullerton, CA, USA) with a 90 cm, 50 µm ID fused-silica capillary (New Objective Littleton, MA, USA) online coupled to a MicroTOF mass spectrometer (BrukerDaltonic, Bremen, Germany). A solution of 20% acetonitrile (Sigma-Aldrich, Taufkirchen, Germany) in HPLC-grade water (Merc, Darmstadt, Germany) supplemented with 0.94% formic acid (Merc KGaA, Darmstadt, Germany) was used as running buffer. The electrospray ionization interface (ESI) sprayer (Agilent Technologies, Palo Alto, CA, USA) was grounded, and the ion spray interface potential was set between −4 and −4.5 kV. Spectra were accumulated every 3 s over a range of mass-to-charge from 350 to 3000. The sample acquisition time was 60 min.

The obtained CE-MS spectra were analyzed using MosaFinder software [1]. Only signals observed in a minimum of three consecutive spectra with a signal-to-noise ratio >3 were considered. Internal standards as reference for mass and migration time by applying global and local linear regression were used for data calibration. The obtained peak list of each polypeptide is characterized by molecular mass, CE-migration time, and normalized ion signal intensity. Signal intensities were used as a measure of relative abundance and normalized using 29 internal standard peptides [19]. All detected peptides were deposited, matched, and annotated in a Microsoft SQL database, permitting further correlation and statistical analysis.

Raw data from the CE-MS analysis from 435 urine samples of patients used in this study are available at Zenodo (https://zenodo.org/record/4964524, accessed on 9 July 2021).

### 2.3. Sequencing of Peptides

Urinary peptides were sequenced using CE- tandem mass spectrometry (MS/MS) or liquid chromatography (LC)-MS/MS, as described in detail [20]. MS/MS experiments were performed using an Ultimate 3000 nano-flow system (Dionex/LC Packings, Sigma-Aldrich, Taufkirchen, Germany) or a P/ACE MDQ capillary electrophoresis system (Beckman Coulter, Fullerton, CA, USA), both connected to an LTQ Orbitrap hybrid mass spectrometer (Thermo Fisher Scientific Inc., Waltham, MA, USA) equipped with a nano-electrospray ion source. The mass spectrometer is operated in data-dependent mode to automatically switch between MS and MS/MS acquisition. Survey full-scan MS spectra (from *m/z* 300–2000) were acquired in the Orbitrap. Ions were sequentially isolated for fragmentation. Data files were searched against the UniProt human nonredundant database using Proteome Discoverer 2.4 and the SEQUEST search engine without enzyme specificity (activation type: HCD; precursor mass tolerance: 5 ppm; fragment mass tolerance: 0.05 Da). No fixed modifications were selected, and oxidation of methionine and proline were selected as variable modifications. The minimum precursor mass was set to 790 Da and maximum precursor mass to 6000 Da with a minimum peak count of 10. For further validation of obtained peptide identifications, the correlation between peptide charge at the working pH of 2 and CE-migration time was utilized to minimize false-positive identification rates [21]. Here, the calculated CE-migration time of the sequence candidate, based on the number of basic amino acids with the sequence, was compared to the experimental migration time.

### 2.4. Protease Prediction

The open-source tool for protease prediction Proteasix (www.proteasix.org, accessed on 25 June 2021) was used in order to link urinary peptides to the proteases potentially involved in their generation [22]. Proteasix uses information about naturally occurring peptides, that is, the corresponding protein UniProt identifier and start/stop amino acid position to predict potential cleaving proteases. Only proteases observed to match cleavage site associations retrieved from the literature were considered (“observed mode”). A list of predicted proteases was generated as a result of the analysis.

### 2.5. Statistical Methods

For the definition of biomarkers, the statistical analysis was performed using R-based statistic software. Only peptides with available amino acid sequence (*n* = 4080) were used in statistical analysis. In addition, a peptide frequency threshold of at least 30% in one of the groups was considered. Wilcoxon rank sum test was used for the calculation of the *p*-values. The *p*-values were adjusted for multiple testing assessed by the method described by Benjamini and Hochberg [23]. Potential biomarkers were combined in a support vector machine (SVM)-based classifier.

The non-parametric Spearman’s rank correlation analyses were performed using MedCalc software (version 12.1.0.0; MedCalc Sofware, Mariakerke, Belgium).

Receiver-operating-characteristic (ROC) curves [24,25] were generated for the classification of the patient samples with the classifier. The ROC curve was obtained by plotting all sensitivity values (true positive fraction) on the y axis against their equivalent (1-specificity) values (false positive fraction) on the x axis for all available thresholds. Each point on the ROC plot represents a sensitivity/specificity pair corresponding to a particular decision threshold. The area under the ROC curve (AUC) was evaluated as it provides a single measure of overall accuracy independent of any threshold. Calculation of 95% confidence intervals (Cl) was based on exact binomial calculations, and the optimal balance of sensitivity and specificity was determined based on the Youden index J. For the ROC analysis, the MedCalc software was used as well.

Cohort matching for eGFR, proteinuria, age, and sex was performed using nearest neighbor interpolation in R-based software.

## 3. Results

### 3.1. Patient Characteristics

Four-hundred-and-thirty-five patients with a renal biopsy and an assessment of IFTA by a renal pathologist were enrolled in this study. The patients’ characteristics are given in Table 1.

The following primary histological diagnoses were included as singular patient groups: acute tubular necrosis (ATN, *n* = 18, including one Crush nephropathy), myeloma cast nephropathy (CAST, *n* = 12), diabetic nephropathy with nodular nephrosclerosis (DNP, *n* = 23), primary focal segmental glomerulosclerosis (FSGSp, *n* = 19), hypertensive ischemic nephropathy (HINP, *n* = 86), IgA nephropathy (IGANP, *n* = 84), Henoch-Schönlein purpura (IGAPSH, *n* = 12), interstitial nephritis (INTN, *n* = 24), lupus nephritis (LN, *n* = 15), minimal change glomerulopathy (MCGN, *n* = 14), membranous nephropathy (MEMGN, *n* = 25), and vasculitis (VASCulitis, *n* = 41). The following primary diagnoses were grouped according to similar pathogenetical mechanisms: paraprotein-associated diseases (AMYLOID, including amyloidosis, fibrillary glomerulonephritis and light chain deposit disease, *n* = 9), membranoproliferative GN-like diseases (C3MPPI_GP, including membranoproliferative glomerulopathy, C3 glomerulopathy and post-infectious glomerulonephritis, *n* = 18), collagen IV-associated diseases (COLIVAD, including Alport′s syndrome and thin basement membrane disease, *n* = 13), and pathologies associated with vascular occlusion (VASCular, including thrombotic microangiopathy, renal cholesterol atheroemboli, and ischemic glomerulopathy, *n* = 12). We also included five cases of secondary FSGS without clear primary cause and five CKD probe sets without clear histopathological diagnosis.

### 3.2. Relationship between IFTA and Clinical Parameters

IFTA assessed in percentage of fibrosis of interstitial area was correlated to eGFR, proteinuria, and age. In fact, the percentage of IFTA showed moderate, but statistically significant correlation with patient eGFR, proteinuria levels, and patient age as shown in Figure 1a–c.

### 3.3. Definition of Urinary Peptides Associated with IFTA

For the definition of peptides associated with IFTA, we generated a training cohort based on two patient groups matched for eGFR, proteinuria, age, and sex. The two groups were separated according to the percentage of IFTA seen in the kidney biopsy. Patients with IFTA ≥10% and <15% were not used for biomarker definition (*n* = 49). The final training cohort included 100 non-fibrosis and 100 fibrosis patients.

After matching of the two groups of the training cohort, no statistically significant differences (Figure 1d–f)) between these groups regarding eGFR (*p* = 0.1252), proteinuria (*p* = 0.1551), age (*p* = 0.5816), and sex (*p* = 0.2491, no graph shown) were observed.

For the definition of IFTA-associated biomarker, the CE-MS data of the training cohort was used. Only peptides with a frequency >30% in at least one of the groups were considered in the statistical comparison between the IFTA <10% and IFTA ≥15% group. We determined a total of 243 peptides with significant differences between the fibrosis and non-fibrosis groups (Wilcoxon rank sum test test, *p* < 0.05). Following adjustment by Benjamini and Hochberg, we obtained 29 still significant urinary peptides. These were combined using the SVM algorithm to a classifier called fibrosis peptide profile FPP_29BH. These 29 peptides corresponded to a total of 13 different proteins. Nineteen of the 29 peptide fragments corresponded to seven different collagen chains.

The classifier peptides and their properties are listed in Table 2.

Furthermore, the 100/100 training cohorts matched for eGFR, proteinuria, sex, and age were subsequently matched for CKD etiology. This matching resulted in a smaller cohort of 55 fibrosis and 55 no-fibrosis patients. The number of patients for each CKD etiology and each fibrosis group was (for abbreviations see methods): AMYLOID 2, ATN 1, C3MPPI_GP 3, CAST 2; COLIVAD 3, DNP 2, FSGSp 4, HINP 6; IGANP 12; IGAPSH 1; INTN 4; LN 2; MEMGN 3; VASCular 1; VASCulitis 9.

Similar to the 100/100 matched cohorts, in the 55/55 CKD etiology-matched cohorts there was no difference between the two groups regarding eGFR, proteinuria, age, and sex (data not shown).

Using the aforementioned statistical analysis, 106 IFTA peptides were identified using Wilcoxon rank sum testing. Though, after adjustment by Benjamini and Hochberg for multiple testing, no significant peptide remained. Nevertheless, only one of the 29 peptides used in the classifier showed significant opposite regulation in the etiology-matched smaller cohorts (Table 2).

### 3.4. Validation of the FPP_29BH Classifier

The FPP_29BH classifier containing 29 specific fibrosis biomarkers was first validated using the cross-validated training data by application of the take one out procedure (100/100 cohort matched for eGFR, proteinuria, age, and sex). The FPP_BH29 resulted in an AUC of 0.851 and 95% CI in the range of 0.800 to 0.902 (*p* < 0.0001) on the no fibrosis (IFTA < 10%) and fibrosis (IFTA ≥ 15%) patients in ROC. The ROC curve is presented in Figure 3.

The FPP_BH29 classifier was then validated using independent samples that had not previously been used in the 100/100 matched cohort for biomarker identification and classifier generation (test set, *n* = 235). The patients with IFTA between 10 and 15% were excluded (*n* = 49) and 40 patients were defined as non-fibrotic (IFTA < 10%) and 146 as fibrotic (IFTA ≥ 15%). The application of the FPP_BH29 on this test set resulted in AUC of 0.840 (95% CI: 0.779 to 0.889, Figure 2). Applying the optimal classification threshold based on the Youden index at 0.025, classification of this independent test cohort resulted in a sensitivity of 74.0% (95% CI: 66.1–80.9) and a specificity of 90.0% (95% CI: 76.3–97.2).

Furthermore, the classifier was applied to all test set data (*n* = 235) with inclusion of patients with an IFTA percentage between 10 and 15%. Figure 3 shows a highly significant correlation between IFTA and the FPP_BH29 classifier with a Rho value of 0.496 (*p* < 0.0001).

### 3.5. Prediction of Proteases

The prediction of protease involved in the generation of the 29 significant peptides resulted in 12 proteases that had at least one protease/cleavage site association reported in the literature. All proteases are listed in Table 3. Proteases with more than one protease/cleavage site association are cathepsin D (CTSD), 72 kDa type IV collagenase (MMP2), collagenase 3 (MMP13), and Matrix metalloproteinase-14 (MMP14).

## 4. Discussion

Renal fibrosis is a dynamic process that occurs in almost all progressive CKD and indicates the path towards ESKD [26]. At present, the amount of IFTA can only be assessed by an invasive kidney biopsy with limitations due to histological scoring standards and clinical applicability. Due to its invasive character, its contraindications and possible complications, percutaneous kidney biopsy is usually only performed once for diagnostic purposes and repeat follow-up biopsies are rarely done. In this study, we aimed to supplement kidney biopsy for determination of the degree of IFTA in CKD patients using urinary peptidomics based on CE-MS, allowing even repeated assessments during the disease course.

Peptides and low molecular weight proteins were chosen as targets for investigation for a multitude of reasons: (1) it is not possible to routinely, reproducibly, and comprehensively analyze full proteins (including PTMs). Such analysis requires tryptic digests, inevitably resulting in the introduction of additional variability and loss of information. (2) Peptides and low molecular weight proteins are present as a result of glomerular filtration also in the urine of healthy individuals. As such, a “normal healthy” urine peptidome can be established and used as reference. In contrast to larger proteins, peptides and low molecular weight proteins are not, or only to a minor degree affected by proteinuria, which, in the case of larger proteins, is a very powerful confounder. (3) We hypothesized that kidney, but also any systemic disease, would be initiated and mandate significant and distinct changes in specific peptides and proteins. While the changes in proteins may be very challenging to assess reproducibly, changes in peptides should be detectable, due to the ability to analyze the whole peptide without any manipulation/derivatization, like tryptic digest or specific labelling.

We identified a fibrosis classifier containing 29 peptide fragments of 13 different proteins. The fibrosis classifier was generated using two patient cohorts with IFTA < 10% or ≥15%, matched for sex, age, proteinuria, and eGFR, and was also applied to independent test samples. The fibrosis classifier was highly significantly associated with IFTA and able to distinguish high from low renal fibrosis in an independent test set.

As expected, we found significant correlation between IFTA and several other patient parameters including age, proteinuria, and most significantly eGFR. This finding was not previously observed in a smaller cohort when correlating these parameters with histologically determined levels of fibrosis [3]. We speculate that this difference might be due to different disease etiologies. Our cohort had a high number of patients with diabetic nephropathy, IgA nephropathy, and hypertensive ischemic nephropathy. In contrast, the small cohort of Magalhães et al. [3] consisted of 9/42 cases of IgA nephropathy, 1/42 chronic hypertensive nephropathy, and 0/42 diabetic nephropathy. Despite the drawback of a small patient cohort, Magalhães et al. defined seven biomarkers associated with fibrosis. In our training cohort, six of these fibrosis biomarkers showed the same regulation. Moreover, four of them were significant in unadjusted statistic (data not shown).

Our peptide-based classifier was generated by statistical analysis of mass spectrometry training data of 200 patients with and without IFTA. As shown in Table 2, peptides with differential peptide intensities between fibrosis and non-fibrosis with statistical significance led to the inclusion of 29 independent peptides with equal statistical significance for our statistical model. In our model, all 29 peptides are of equal importance for classification of IFTA. In the following, we aim to discuss single differentially regulated peptides and possible underlying pathophysiological mechanisms for differences between the fibrosis and non-fibrosis group.

Alpha-2-HS-glycoprotein (AHSG), also known as fetuin-A, is a plasma binding protein. In our study, four different peptide fragments of fetuin-A were implemented in the fibrosis classifier containing 29 urinary peptides. As seen in Table 2, urinary samples of individuals with high percentage of IFTA showed an increased amount of urinary fetuin-A. In accordance with our data, Schanstra et al. also described a negative correlation between AHSG and baseline eGFR in a large cohort of over 500 patients [5].

Furthermore, peptide fragments of AHSG also form part of the CKD273 classifier for CKD, which has been integrated in numerous studies so far and is now commercially available after a letter of approval from the FDA [27,28,29,30]. Urinary AHSG levels have also recently been associated with CKD and negatively correlated with eGFR slope and baseline eGFR [31].

In this study, we did not correlate peptide changes with eGFR of patients. We saw, however, a negative correlation between eGFR and IFTA, which was highly significant. Our findings of a positive correlation between increased AHSG levels in the urine with higher percentages of IFTA seem to be in accord with these previous studies. Fetuin-A is an inflammation-regulated protein involved in regulation of extraosseous calcification via regulation of calcium and an inhibitor of calcification. Intravascular calcification strongly affects cardiovascular mortality, especially in CKD patients [32]. Serum AHSG levels have shown to be significantly lower in patients receiving hemodialysis than in healthy individuals. The sera showed impaired inhibition of CaxPO4 precipitation, which is believed to be one of the reasons for increased cardiovascular calcification [33].

In a meta-analysis among 5169 CKD patients, low serum levels of fetuin-A were associated with increased mortality independent of diabetes and inflammation in dialysis patients [34].

To our knowledge, no clear mechanism that leads to decreased AHSG levels in serum of CKD patients and especially hemodialysis patients has been discovered. Explanations remain speculative and include possible renal post-translational modifications, which are impaired in CKD or increased urinary loss of AHSG. The findings of increased urinary AHSG in patients with fibrotic kidneys in our study could support this hypothesis, even though further investigation of molecular mechanisms is needed.

In our study, 19 different collagen peptide fragments of eight different collagen chains were found with differential intensities between patients with high and low degree of IFTA (for detail see Table 2). For 13 of these fragments, a positive correlation with degree of IFTA was noticed. Six peptides were inversely correlated with IFTA, three of which corresponded to Collagen alpha-1 (I) chain (COL1A1).

Collagens are a key part of the extracellular matrix of the kidney that confer structural integrity, cell adhesion, and serve in various signaling pathways [35]. Renal fibrosis is characterized by an imbalance between formation and degradation of extracellular matrix proteins such as collagens [36].

In previous studies, controversial findings were published regarding urinary collagen loss in CKD and more specifically in renal fibrosis. In a recent study, Magalhães et al. found a negative correlation of urinary collagen fragments and interstitial fibrosis [3]. Regarding the CKD273 classifier, collagen fragments generally are reduced in CKD because of inflammation-driven inhibition of matrix metalloproteinases, which mediate collagen cleavage and thus shift the balance towards collagen formation and away from degradation, apparently resulting in decreased urinary collagen peptides [30]. Our findings of decreased urinary collagen fragments corresponding to COL1A1 are in accord with this hypothesis. As collagens play a central role in the mechanism of disease development of renal interstitial fibrosis in the context of chronic kidney disease, it is not surprising that they are highly present within our peptide-based classifier [12].

Most of the other collagen fragments in our study, however, were upregulated in fibrotic kidney disease. Several studies have shown increased urinary collagen fragments in CKD and could associate this with renal fibrosis [37,38]. Specifically, increased collagen 3 fragments have been well described in animal CKD models and renal fibrosis [38,39]. The consensus is that collagen, as the predominant extracellular matrix (ECM) molecule, plays an important role in renal fibrosis. How single urinary collagen fragments reflect the degree of fibrosis is yet to be understood. Moreover, the turnover of renal collagen seems to be central in fibrosis [36]. Therefore, urinary peptide analysis remains observational and underlying molecular mechanisms like proteolytic events must be studied in order to fully interpret changes in urinary collagen peptides.

Another generally prevalent peptide was the alpha chain of fibrinogen (Table 2). Peptide intensities were elevated in the fibrosis group when compared to no fibrosis. Urinary fibrinogen was recently shown to be an independent risk factor and predictor for CKD [40]. Serum levels of fibrinogen were identified as an independent predictor of mortality in stage 3 and 4 CKD patients [41]. In an animal model, pharmacological and genetic intervention were successfully used to protect kidneys from fibrosis [42]. So, the abundance of fibrinogen in our general cohort as well as the increased abundance in the fibrosis group seems to be in accord with those findings. Similarly, we found increased levels of antithrombin III in urine samples of fibrotic kidneys. Several groups independently reported amelioration of renal ischemia-reperfusion injuries in rats by treatment with antithrombin III [43,44]. These findings, together with our findings, fit very well into the general assumption that procoagulatory processes play an essential role in development and progression of CKD through renal fibrosis [45].

In addition, we saw two peptide fragments of the polymeric immunoglobulin receptor upregulated in the fibrosis group. The polymeric immunoglobulin receptor is a transmembrane protein of mucosal epithelia and has recently been localized in the proximal tubules and parietal epithelial cells of glomeruli in the human kidney [46]. Krawczyk et al. [46] also linked this protein to increased levels of secretory IgA in kidney disease and among others correlated it to the degree of interstitial fibrosis. We and others have also linked urinary polymeric immunoglobulin receptor fragments to severity of kidney injury in patients with CKD in the context of cardio-renal syndrome and IgA nephropathy [47,48].

The finding that two hemoglobin subunits fragments are present in a much larger abundance in non-fibrotic urine samples is not surprising in our view. We suggest that most of CKD patients in advanced stages with higher degrees of renal fibrosis do not normally suffer from hematuria. In contrast, hematuria is more likely seen as a sign of acute inflammation and low degrees of renal fibrosis are to be expected. Nevertheless, there has been a call for hematuria to be integrated as a prognostic factor for CKD [49]. Our data does not currently support this theory. However, we suggest looking at hematuria or urinary hemoglobin peptide fragments as a risk factor in selected pathologies where they might have prognostic value for CKD. For example, Coppo et al. showed a prognostic value of persistent microhematuria for IgA progression [50]. Probably due to our very heterogenous cohort, we see a negative correlation of hemoglobin fragments with IFTA.

Based on the defined 29 fibrosis-associated peptides, we were able to predict proteases probably involved in the generation of these peptides. In general, proteases are regulated on a posttranslational level, and in some cases (e.g., MMPs) also by specific inhibitors (e.g., TIMPs). Consequently, transcriptome data can generally not be used to predict protease activity. Assessment of protease activity is consequently quite challenging, and very little information on this topic is available. We found only one manuscript that investigated the activity of one of the proteases predicted to be deregulated based on our results: CTSD [51]. The authors showed that inhibition of CTSD with pepstatin A in an animal model reduced fibrosis, which is in contrast to the prediction based on urinary peptides. However, it is important to keep in mind that animal models may not well reflect human disease: In a previous publication, we demonstrated that a widely used animal model for human diabetic kidney disease, the ZDF rat, generally shows regulation of urinary collagen fragments opposite to human. The fact that potential efficacy of pepstatin A in human was never reported even though the report in animals was published 7 years ago may indicate the animal data reported by Fox et al. does not reflect human disease.

One of the most striking findings of this study is the highly significant correlation of IFTA and eGFR (Figure 1a). When applying ROC-analysis to independent samples not used in classifier generation, an AUC of 0.987 is obtained. Our data show that around IFTA values of 10–15%, eGFR of our patient cohort drops dramatically, which gives eGFR a strong predictive value when differentiating between the two groups (IFTA ≥ 15% and IFTA ≤ 10%). We see, however, almost no incremental correlation of eGFR at IFTA values > 20%. In contrast, the proteomics classifier FPP_BH29 shows significant correlation throughout IFTA percentages. Therefore, we believe that the fibrosis classifier can add significant value to diagnostical assessment of IFTA and it would be counterintuitive as well as fatal to rely on eGFR as a singular diagnostic tool to assess fibrosis.

## 5. Conclusions

In conclusion, our study could identify a novel proteomics classifier containing 29 urinary peptide fragments reflecting the degree of interstitial fibrosis and tubular atrophy in CKD patients. This gives us a tool to assess the degree of fibrosis during the course of the disease by repeated measurements and thus enables us to better predict prognosis. We also observed highly significant correlation of IFTA with eGFR, proteinuria, and age. We therefore suggest combining routinely assessed markers like eGFR, age, and proteinuria with the newly discovered urinary peptide-based fibrosis classifier FPP_BH29 to evaluate renal fibrosis in addition to kidney biopsy.

## Figures and Tables

**Figure 1 proteomes-09-00032-f001:**
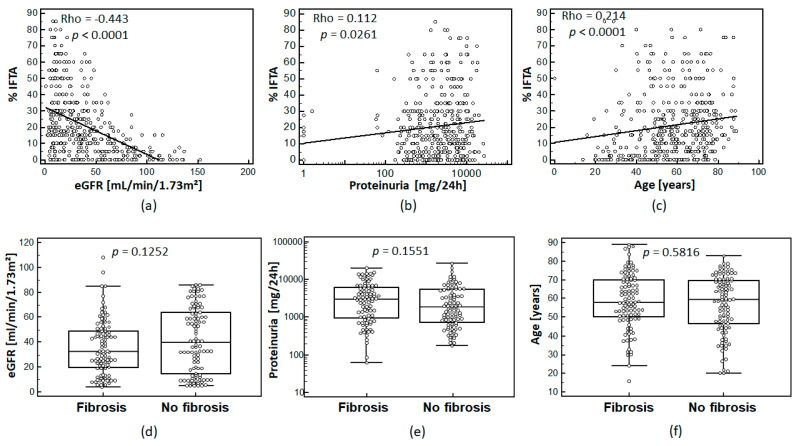
Association of IFTA for the whole cohort (*n* = 435) with eGFR (CKD-EPI) (**a**); proteinuria (**b)**, (logarithmic scale); and age (**c**): Spearman’s coefficient of rank correlation (Rho) and significance level are given on the top left corner of each graph. The distribution of eGFR (**d**); proteinuria (**e**), (logarithmic scale); and age (**f**) in the 100/100 matched cohort for eGFR, proteinuria, and age are shown for matched cohorts. *p*-values between the sub-cohorts are given above the plots.

**Figure 2 proteomes-09-00032-f002:**
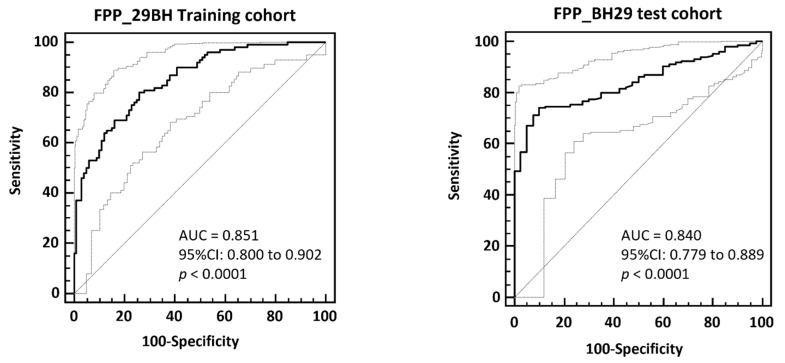
ROC-analysis of FPP_BH29 classifier applied to total cross-validated training data (**left**) and to an independent test set composed of patients with IFTA < 10% and IFTA ≥ 15 (**right**). In the bottom right corner of the graph area under the ROC curve (AUC), 95% confidence intervals and significance levels (*p* < 0.001) are given.

**Figure 3 proteomes-09-00032-f003:**
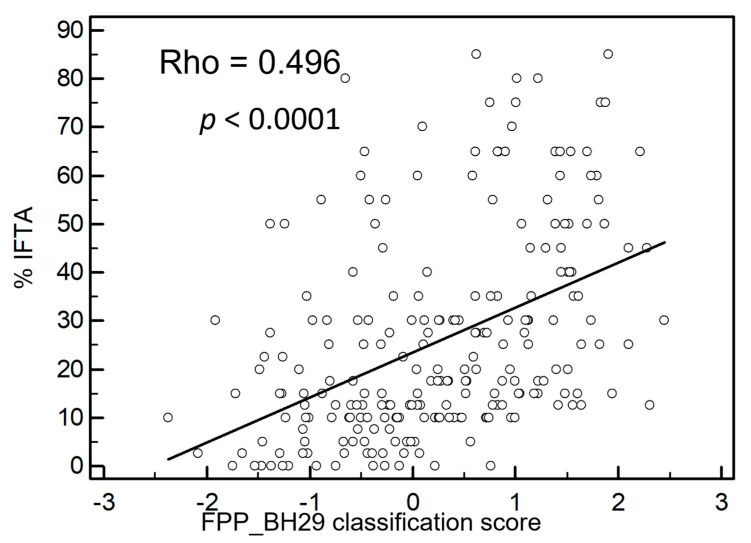
Correlation of IFTA with fibrosis classifier FPP_ BH29. The fibrosis classifier shows a positive correlation with IFTA percentage with a Rho-value of 0.496. For this graph, all independent samples not used for classifier generation were used including samples with IFTA percentages between 10 and 15%; *n* = 235, *p* < 0.0001.

**Table 1 proteomes-09-00032-t001:** Patient characteristics of study cohorts. * Values are given as mean ± SE. Training cohort with two matched sub-cohorts (matching described in methods) with fibrosis and no fibrosis. *p*-values between training and test cohort are given in right column (Student’s *t*-test).

Study Cohort (*n* = 435)	Training Cohort		Test Cohort	*p* Test VS. Training
Characteristics	Fibrosis	No Fibrosis		
Number of subjects	100	100	235	
Age (years) *	58.8 ± 14.9	56.9 ± 15.4	58.3 ± 17.5	0.793
Gender (men/women)	64/36	55/45	139/96	*n*/a
eGFR (mL/min/1.73 m^2^, CKD-EPI) *	35.9 ± 21.8	41.6 ± 26	37.7 ± 35.3	0.714
Proteinuria (mg/24 h) *	4274 ± 4415	3809 ± 4395	3785 ± 4074	0.814
IFTA% *	29.1 ± 13.9	2.6 ± 2.8	25.7 ± 21.1	1.267 × 10^−7^

**Table 2 proteomes-09-00032-t002:** Twenty-nine defined biomarkers for fibrosis: In the first column, gene symbols are given followed by amino acid sequences. Also listed are the adjusted *p*-values; mean peptide intensities (normalized using internal standards) in fibrosis (IFTA ≥ 15%) and no fibrosis (IFTA < 10%) groups are given with fold change (fibrosis/no fibrosis) for the training cohort 100/100. In addition, the unadjusted Wilcoxon *p*-values, mean intensities, and fold change for the etiology-matched cohort are given (55/55). * This peptide could not be validated in the etiology-matched cohort (significant regulation in opposite direction).

Urinary Peptides	Training Cohort 100/100					Etiology Matched Cohort 55/55			
Gene Symbol	Sequence	adj. *p*-Value (BH)	Mean Intensity Fibrosis	Mean Intensity No Fibrosis	Fold Change	Unaj.Wilcox-*p*-Value	Mean Intensity Fibrosis	Mean No Fibrosis	Fold Change
COL10A1	GHPGPSGPPGKpGYGSpGLQGEpGLPGPPGPS	2.42 × 10^−4^	782.77	380.77	2.056	1.34 × 10^−2^	755.71	443.15	1.705
COL1A2	GPQGVQGGKGEQGPPGPPGFQGLPGPSGpAGEVGKpGERG	2.42 × 10^−4^	1151.22	272.13	4.230	3.21 × 10^−3^	1108.09	328.9	3.369
COL1A2	DQGPVGRTGEVGAVGpPGFAGEKGPSGEAGTAGPpGTpGP	8.56 × 10^−4^	196.08	75.14	2.610	1.22 × 10^−2^	168.37	83.85	2.008
AHSG	SLGSPSGEVSHPRKT	8.72 × 10^−4^	2282.82	975.54	2.340	4.44 × 10^−4^	2515.71	1480.13	1.7
AHSG	VVSLGSPSGEVSHPRKT	9.37 × 10^−3^	11,404.37	7829.98	1.457	3.12 × 10^−2^	13,131.49	12,697.43	1.034
PIGR	LFAEEKAVADTRDQADGSRASVDSGSSEEQGGSSRA	1.22 × 10^−2^	774.23	654.63	1.183	2.49 × 10^−3^	624.32	360.05	1.734
COL1A2	VGRTGEVGAVGPpGFAGEKGPSGEAGTAGPpGTpGP	1.82 × 10^−2^	109.41	46.41	2.357	2.91 × 10^−1^	92.33	59.3	1.557
COL3A1	ARGLpGppGSNGNPGPPGPSGSPGKDGPPGPAGNTGAPG	2.34 × 10^−2^	902.6	594.85	1.517	1.01 × 10^−1^	889.74	771.39	1.153
SERPINC1	FSPEKSKLPGIVAEGRDDLYVSDAFHKAF	2.34 × 10^−2^	9438.98	5838.02	1.617	1.80 × 10^−3^	11,768.37	8101.28	1.453
COL2A1	GETGAAGpPGpAGPAGERGEQGAPGP	2.34 × 10^−2^	43.02	135	0.319	1.03 × 10^−2^	58.08	160.81	0.361
COL4A1	pGIPGFPGSKGEMGVMGTPGQPGSPGPVGAPGLPGEKGDH	2.34 × 10^−2^	3045.2	1456.47	2.091	4.89 × 10^−2^	2563.4	1629	1.574
COL1A1	ANGApGNDGAKGDAGApGApGSQGApGLQGMpGERGAAGLPGp	2.69 × 10^−2^	1210.38	814.9	1.485	1.94 × 10^−1^	1267.81	1012.08	1.253
COL3A1	ApGPAGSRGApGPQGpRGDKGETGERG	2.69 × 10^−2^	1103.52	618.43	1.784	1.27 × 10^−1^	857.9	692.61	1.239
COL1A1	GADGQPGAKGEpGDAGAKGDAGPpGPAGP	2.69 × 10^−2^	108.1	388.09	0.279	7.21 × 10^−3^	106.07	519.75	0.204
COL1A1	ANGApGNDGAKGDAGApGApGSQGApGLQGMpGERGAAGLpGp	2.74 × 10^−2^	447.16	170.81	2.618	2.04 × 10^−1^	470.53	214.82	2.19
HBA1	AAHLPAEFTPAVHASLDKFL	2.81 × 10^−2^	610.19	14,067.56	0.043	9.65 × 10^−3^	903.23	19,261.46	0.047
COL1A1	ADGQpGAKGEpGDAGAKGDAGPPGPAGP	2.81 × 10^−2^	212.86	365.65	0.582	1.01 × 10^−1^	217.32	302.79	0.718
COL3A1	EGGKGAAGpPGPpGAAGTpGLQG	2.81 × 10^−2^	689.84	500.7	1.378	7.94 × 10^−2^	705.78	551.83	1.279
COL22A1	GTEGKKGEAGPPGLPGPpGIAGpQGSQGERGADGEVGQKGDQGHPGVPGFMGPPGNPGP	2.81 × 10^−2^	192.19	159.22	1.207	4.74 × 10^−2^	171.5	166.83	1.028
AHSG	GVVSLGSPSGEVSHPRKT	2.81 × 10^−2^	2476.64	1429.55	1.732	2.24 × 10^−2^	2876.18	2229.17	1.29
PIGR	FAEEKAVADTRDQADGSRASVDSGSSEEQGGSSRALVSTLVPL	3.06 × 10^−2^	891.67	377.71	2.361	3.43 × 10^−2^	861.08	320.08	2.69
COL2A1	ppGSNGNpGPPGPPGPSGKDGPKGARGDSGPPGRAGEPG	3.50 × 10^−2^	412.14	184.53	2.233	2.54 × 10^−1^	396.08	243.4	1.627
COL18A1	DDILASPPRLPEPQPYPGAPHHSS	3.77 × 10^−2^	611.67	433.29	1.412	5.12 × 10^−1^	560.24	524.1	1.069
COL3A1	EpGRDGVpGGPGm	3.77 × 10^−2^	2254.17	1608.12	1.402	1.08 × 10^−2^	2160.1	1392.93	1.551
HBA1	AAHLPAEFTPAVHASLDKFLAS	4.15 × 10^−2^	847.52	30,583.66	0.028	3.06 × 10^−2^	1046.13	30,595.91	0.034
FGA	DEAGSEADHEGTHSTKRGHAKSRPV	4.15 × 10^−2^	31,926.35	22,421.31	1.424	5.01 × 10^−1^	29,485.54	34,133.85	0.864
AHSG	VSLGSPSGEVSHPRKT	4.15 × 10^−2^	3680.2	2187.3	1.683	2.25 × 10^−2^ *	2326.63	3525.8	0.66 *
COL3A1	GpGSDGKPGPpG	4.86 × 10^−2^	145.26	347.21	0.418	2.28 × 10^−2^	192.46	399.37	0.482
COL1A1	GSpGSpGPDGKTGPPGPAG	4.86 × 10^−2^	74.29	178.76	0.416	4.19 × 10^−2^	68.72	157.74	0.436

**Table 3 proteomes-09-00032-t003:** List of predicted proteases involved in the generation of the 29 fibrosis-associated peptides. Cleavage proteins, predicted involved proteases, and number of associated cleavage events (↓ down and ↑ regulated) are indicated. Given is also the fold change (average fibrosis/average no fibrosis based on training data) and the *p*-value (Mann-Whitney). Bold marked are proteases with number of cleavage events >1.

Cleaved Proteins	Protease (Gene)	*n* of Cleaving Sites		Fold Change	Average Fibrosis	Average No Fibrosis	*p*
		↓	↑				
HBA1 (5)	**Cathepsin D (CTSD)**	5	0	0.03	752.59	23977.22	0.0006
COL2A1 (1)	Macrophage metalloelastase (MMP12), Neutrophil collagenase (MMP8)	1	0	0.32	43.02	135.00	0.0002
COL1A1 (1), COL1A2 (2)	**72 kDa type IV collagenase (MMP2)**	1	2	1.26	126.59	100.10	0.1580
COL2A1 (1), COL18A1 (2), COL1A2 (2)	**Collagenase 3 (MMP13)**	1	4	1.40	314.37	224.63	0.0002
COL18A1 (1)	Cathepsin B (CTSB), Cathepsin K (CTSK), Procathepsin L (CTSL), Matrix metalloproteinase-20 (MMP20), Matrilysin (MMP7)	0	1	1.41	611.67	433.29	0.0006
COL18A1 (1), COL1A2 (2)	**Matrix metalloproteinase-14 (MMP14)**	0	3	1.65	305.72	184.95	0.0001
COL2A1 (1)	Interstitial collagenase (MMP1)	0	1	2.23	412.14	184.53	0.0006

## Data Availability

Mass spectrometry data (CE-MS) of 435 analyzed samples used in this study are deposited at Zenodo (https://zenodo.org/record/4964524, accessed on 9 July 2021).

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
