# Peer review of "A Novel Urinary Proteomics Classifier for Non-Invasive Evaluation of Interstitial Fibrosis and Tubular Atrophy in Chronic Kidney Disease"

_proteomes, 2021, doi:10.3390/proteomes9030032_

Round 1
Reviewer 1 Report
Dear Authors,
The article “A novel urinary proteomics classifier for non-invasive evaluation of interstitial fibrosis and tubular atrophy in chronic kidney disease” is interesting from the scientific point of view and it reports a potential novel diagnostic strategy related to non-invasive urinary peptides biomarkers in detection and prediction of chronic kidney disease. The Authors presented the experimental results regarding the urinary peptides profiles of 435 patients (proteomics data). The manuscript is written in a clear manner, however some sections and graphs require improvement. Generally, the manuscript requires some minor corrections before acceptance, such as the improvement of the tables, charts and several other aspects listed below. Moreover, the Authors should also check for possible grammar/spelling typos, as they are present in the manuscript.
Minor concerns:
- Fig. 1 and Fig. 2 Graphs are not readable, please enlarge them and provide more readable charts, as their quality is inadequate.
- Please insert subscripts in chemical formulas where necessary (e.g. in the line 115 and 119).
- Material and methods section should be improved. For every kit, reagent and chemical cited in Material and methods, Authors need to detail the name of the vendor, the city of the vendor and its country (e.g. for Sigma Aldrich in line 124). In the case of USA cities, also the State. Example: Sigma-Aldrich Merck, Sant Louis, MO, USA.
- Lines 265 and 266 are grammatically incorrect.
- Table 2 is unreadable, please make it more "reader friendly" and see examples of tables in the manuscripts published in MDPI journals.
- Please put the spaces between ±, < and values in all the manuscript
Kind regards.
Author Response
Review 1:
Dear Authors,
The article “A novel urinary proteomics classifier for non-invasive evaluation of interstitial fibrosis and tubular atrophy in chronic kidney disease” is interesting from the scientific point of view and it reports a potential novel diagnostic strategy related to non-invasive urinary peptides biomarkers in detection and prediction of chronic kidney disease. The Authors presented the experimental results regarding the urinary peptides profiles of 435 patients (proteomics data). The manuscript is written in a clear manner; however, some sections and graphs require improvement. Generally, the manuscript requires some minor corrections before acceptance, such as the improvement of the tables, charts and several other aspects listed below. Moreover, the Authors should also check for possible grammar/spelling typos, as they are present in the manuscript.
Minor concerns:
- 1and Fig. 2 Graphs are not readable, please enlarge them and provide more readable charts, as their quality is inadequate.
We have improved the quality of the image and merged images 1 and 2 in accordance with the comment from reviewer 2.
- Please insert subscripts in chemical formulas where necessary (e.g., in the line 115 and 119).
We have checked the manuscript for subscripts in chemical formulas and corrected missing subscripts.
- Material and methods section should be improved. For every kit, reagent and chemical cited in Material and methods, Authors need to detail the name of the vendor, the city of the vendor and its country (e.g., for Sigma Aldrich in line 124). In the case of USA cities, also the State. Example: Sigma-Aldrich Merck, Sant Louis, MO, USA.
According to the comments, we have now thoroughly revised the paper by adding the missing details for every kit and reagent.
- Lines 265 and 266 are grammatically incorrect.
Grammatical mistakes in the mentioned lines have been corrected. Furthermore, a grammar and spell check has been performed by a native English speaker.
- Table 2 is unreadable, please make it more "reader friendly" and see examples of tables in the manuscripts published in MDPI journals.
To make the table 2 more “reader friendly” we removed some columns which were not essential, enlarged the font and adjusted the formation to MDPI tables.
- Please put the spaces between ±, < and values in all the manuscript
The manuscript has been checked for spaces between these signs and revised accordingly.

Reviewer 2 Report
Manuscript is not for publication, it needs major revision
Abstract should be clear and concise with clear concussion what peptide, s is/are crucial for diagnostics for interstitial fibrosis and tubular atrophy
In the Introduction and Discussion elaborate more on chronic kidney disease interstitial fibrosis and tubular atrophy, mechanism of disease development and clinics
Figure 1 Legend needs detailed explanation,
Figure 1 and Fig 2 to be one figure
Table 2 Legend needs detailed explanation with detailed significant peptides for diagnostics of interstitial fibrosis and tubular atrophy
Fig 4 and 5 to be one figure but with detailed figure legend and explanation of FPP –BH29
All abbreviation should have explanation
In the Methods elaborate more on ROC analysis. Elaborate more on capillary electrophoresis coupled with mass spectrometry in relation to clinical applications in urinary peptide profiles
Authors need to show crucial peptide for diagnostics of these presented on table 2 and their correlation to clinical development and clinical symptoms and signs and to be discussed in the Discussion
Author Response
Review 2
- Abstract should be clear and concise with clear concussion what peptide, s is/are crucial for diagnostics for interstitial fibrosis and tubular atrophy.
We hope we understood and interpreted the reviewer comment correctly. As to the point of crucial peptides for diagnosis of IFTA in CKD we identified a peptide classifier containing 29 differentially regulated peptides in urine samples of individuals with fibrotic vs. non-fibrotic kidneys. As they form a classifier, they are in our opinion equally important for non-invasive evaluation of fibrosis. Our aim was not to single out one or few specific peptides that are crucial for diagnostics. We tried to further clarify this fact by alternating the abstract accordingly.
- In the Introduction and Discussion elaborate more on chronic kidney disease interstitial fibrosis and tubular atrophy, mechanism of disease development and clinics.
We thank the reviewer for this suggestion and expanded introduction and discussion focussing more on underlying molecular mechanisms. As for the clinics part we found that our introduction is already focussing a lot on clinical aspects of clinical evaluation of IFTA and CKD as well as clinical implications. We hope we met the expectations of the reviewer with our changes.
- Figure 1 Legend needs detailed explanation.
As suggested by the Reviewer we applied more detailed explanation of figure 1. We generally tried to expand figure legends with more detailed information on the corresponding figure/graph.
- Figure 1 and Fig 2 to be one figure.
As suggested by the reviewer, we have merged images 1 and 2 and improved the quality of the image in accordance with the comment from reviewer 1.
- Table 2 Legend needs detailed explanation with detailed significant peptides for diagnostics of interstitial fibrosis and tubular atrophy.
We hope that we understand and interpret the reviewer comment correctly, that more explanation is needed for Table 2. To address this comment, we have revised the legend, aiming towards a clearer description of the contents of Table 2 *.
- Fig 4 and 5 to be one figure but with detailed figure legend and explanation of FPP –BH29
As suggested, figure legends were expanded and include more detailed information. As for merging figures 4 and 5 we would rather prefer to have figure 5 as a singular graph as it stresses and important point that not only the classifier can distinguish between high and low IFTA but it also correlates with IFTA percentage upon inclusion of previously excluded samples (IFTA between 10 and 15 %).
All abbreviation should have explanation.
We thoroughly checked the manuscript for unexplained abbreviations and made sure that every used abbreviation is explained at least when first used in the paper.
- In the Methods elaborate more on ROC analysis. Elaborate more on capillary electrophoresis coupled with mass spectrometry in relation to clinical applications in urinary peptide profiles
We would like to thank the Reviewer for the comment. To address it, we provided some additional information about the ROC analysis in the methods together with additional references. We also added a short section in the beginning about CE-MS and why this technology was chosen, as well as a reference for clinical application of CE-MS (Tofte et al. 2020). In the light of the extensive description of the technical details especially in Mischak et al. 2013 (REF 16), we felt expanding on the methods would constitute an unnecessary repetition of already published information. We very much hope the reviewer agrees with this approach.
- Authors need to show crucial peptide for diagnostics of these presented on table 2 and their correlation to clinical development and clinical symptoms and signs and to be discussed in the Discussion.
We hope that we understood this comment correctly. As mentioned before, we struggle to identify singular crucial peptides for clinical development and especially symptoms. Our classifier-based approach does not aim to unravel underlying molecular mechanisms but to offer a practical tool for non-invasive evaluation of renal fibrosis. All 29 peptides are important for classifier formation. In the discussion we tried to touch the aspect that collagens are represented in high abundance within the classifier but show differential regulation in fibrosis regarding different chains. As for clinical development and symptoms we are very cautious to discuss specific peptide mechanisms as our cohort is very heterogenous including many different disease etiologies with different symptoms and clinical presentations. We tried to elaborate more on this fact within the discussion and hope to meet the reviewer’s expectation.

Reviewer 3 Report
It is a great honor and pleasure for me to be invited as the reviewer for this interesting work. Catanese et al. evaluated the clinical predictive values of urinary proteomics classifier for prediction of interstitial fibrosis and tubular atrophy (ITFA) in chronic kidney disease (CKD). This study topic is novel, attributing to Prof. Justyna Siwy’s long-term efforts and contributions in this scientific field. Although the article is well-written, I have a number of comments concerning this study:
- Abstract:
Line 31: The main result “(rho= - 0.5, p<0.0001;” should be corrected as “(rho= 0.5, p<0.0001); ” .
- Line 31: The word “conclusion:” should be added after the subheadings (4).
repeat measurement => repeated measurement
- From the perspective of nephrologists, IFTA is not a solid renal outcome. Although the proteomic classifier is novel, the correlations between IFTA and age/eGFR/proteinuria degree are well-known without The aim of the study “non-invasive evaluation of IFTA in CKD” is not practical at all. In clinical practice, renal echo is a not only a non-invasive tool but also reliable measurement for renal size/atrophy/ fibrosis severity. Most of all, renal echo exam is very convenient and inexpensive. The major purpose of the renal biopsy is to confirm etiology (eg., differential diagnosis for GN types), contributing to subsequent therapeutic strategy. Since the proteomic classifier could not replace renal biopsy that renal biopsy remains inevitable, the clinical utility of proteomic classifier is suboptimal. Considering the study period is relatively long (from 2008 to 2020), the authors may test the predictive value for solid renal endpoints (dialysis, halving of GFR or doubling of serum creatinine, etc.). A precise and reliable prediction model for CKD progression is of great importance.
- Table 1: p-values between the training cohort and test cohort are required.
- Why did authors define the exclusion criteria of IFTA 10-15%? There is a diagnostic dilemma for patients with IFTA 10-15%.
- Please provide the information of the conditions of capillary electrophoresis. If the method was developed in previous study, please provide key parameters such as applied voltage, cartridge type (or the specifications of capillary) and running time.
- How the peptides were normalized? If the normalized peptides intensities were used for fold change comparison, please indicate in the table 2.
- In the result of Figure 1, I would like to suggest the authors change the “highly significant association” to “moderate correlation with statistical significance”. The rho values performed here are actually not high correlation.
- Please unify the performance of numbers in table 1 and table 2. For example, is the fold change 2000 folds or 2 folds? Please check the use of “,” and “.” in the numbers.
- Why did the authors choose peptides instead of proteins as investigating targets? Is it because of the saving in sample preparation time?
Thank you for giving me the opportunity to review this interesting article. After major revision, this interesting article could be considered for publication.
Sincerely,
Author Response
Review 3
Abstract:
- Line 31: The main result “(rho= - 0.5, p<0.0001;” should be corrected as “(rho= 0.5, p<0.0001); ”
We corrected the main result and omitted the “-“.
- Line 31: The word “conclusion:” should be added after the subheadings (4).
We would like to thank the reviewer for his thorough reading of our article. The abstract is arguably the most important feature of an article and gives an important first impression. Therefore, we appreciate it a lot that the Reviewer noticed these significant typing mistakes.
2.1 repeat measurement => repeated measurement
We corrected this according to the review.
- From the perspective of nephrologists, IFTA is not a solid renal outcome. Although the proteomic classifier is novel, the correlations between IFTA and age/eGFR/proteinuria degree are well-known without The aim of the study “non-invasive evaluation of IFTA in CKD” is not practical at all. In clinical practice, renal echo is a not only a non-invasive tool but also reliable measurement for renal size/atrophy/ fibrosis severity. Most of all, renal echo exam is very convenient and inexpensive. The major purpose of the renal biopsy is to confirm etiology (eg., differential diagnosis for GN types), contributing to subsequent therapeutic strategy. Since the proteomic classifier could not replace renal biopsy that renal biopsy remains inevitable, the clinical utility of proteomic classifier is suboptimal. Considering the study period is relatively long (from 2008 to 2020), the authors may test the predictive value for solid renal endpoints (dialysis, halving of GFR or doubling of serum creatinine, etc.). A precise and reliable prediction model for CKD progression is of great importance.
We thank the reviewer for this very productive comment. We agree that imaging techniques have been developed and improved in the last two decades for non-invasive evaluation of renal fibrosis including ultrasound- and MRI-based techniques (e.g. elastography). Although, classic renal echo cannot reliably determine IFTA. These diagnostic tools share some aspects with a proteomics based urinary classifier such as cost efficiency, repeatability and low risks due to their non-invasive character. To our knowledge, these techniques have limitations when it comes to very dynamic processes such as fibrotic remodelling and have low sensitivity and specificity especially when it comes to the early stages of fibrosis development where vast eGFR decline is present.
We agree that at the current state kidney biopsy remains pivotal for diagnosis of CKD. Follow-up evaluation though could be realised using a combination of imaging techniques and urinary proteomics.
The aim of our current study was to find a urinary proteomics classifier specifically for evaluation of renal fibrosis as a singular aspect of a corresponding kidney biopsy. Prediction models for CKD using urinary proteomics have previously been published by our colleagues (e.g. Schnastra et al.). Though, we will, as suggested, evaluate our cohort for urinary peptide classifiers regarding classical endpoints.
- Table 1: p-values between the training cohort and test cohort are required.
We thank the reviewer for this comment. P-values between cohorts have been added to check for statistical difference of training and test cohort attributes.
- Why did authors define the exclusion criteria of IFTA 10-15%? There is a diagnostic dilemma for patients with IFTA 10-15%.
In previous studies it became evident that when separation in cases and controls is not possible with certainty (REF IgAN prediction, Covid prediction), exclusion of the "grey area of increased uncertainty", consequently improvement of certainty of assigning group membership (case or control) results in cleaner datasets, leading to improvement in the definition of biomarkers. Based on this observation, we choose to define a grey zone for IFTA, the range of 10-15%, and exclude the associated samples from analysis.
- Please provide the information of the conditions of capillary electrophoresis. If the method was developed in previous study, please provide key parameters such as applied voltage, cartridge type (or the specifications of capillary) and running time.
We would like to thank the Reviewer for the comment, giving an opportunity to indicate the maturity of the technology platform. We provided some more details and references for CE-MS in the methods. We added the information about the acquisition time (60 minutes), capillary used (fused-silica capillary, 90 cm length, 50 µm ID). The information about the applied voltage were already included in the methods section.
- How the peptides were normalized? If the normalized peptides intensities were used for fold change comparison, please indicate in the table 2.
The peptides intensities are normalized using 29 internal internal standard peptides as described in the methods. We added this information also to the legend of table 2.
- In the result of Figure 1, I would like to suggest the authors change the “highly significant association” to “moderate correlation with statistical significance”. The rho values performed here are actually not high correlation.
The wording was changed according to the reviewer’s suggestion. The order of the words was changed to avoid using the word “with” twice in a row which would have caused syntax problems.
- Please unify the performance of numbers in table 1 and table 2. For example, is the fold change 2000 folds or 2 folds? Please check the use of “,” and “.” in the numbers.
The tables were checked and number performance was unified. Values should now be clear.
- Why did the authors choose peptides instead of proteins as investigating targets? Is it because of the saving in sample preparation time?
Peptides and low molecular weight proteins were chosen as targets for investigation for a multitude of reasons: 1) it is not possible to routinely, reproducibly and comprehensively analyse full proteins (including PTMs). Such analysis requires tryptic digests, inevitably resulting in introduction of additional variability, and loss of information. 2) Peptides and low molecular weight proteins are present as a result of glomerular filtration also in the urine of healthy individuals. As such, a "normal healthy" urine peptidome can be established and used as reference. In contrast to larger proteins, peptides and low molecular weight proteins are not or only to a minor degree affected by proteinuria, which, in the case of larger proteins, is a very powerful confounder. 3) We hypothesized that kidney disease, but also any systemic disease would cause significant and distinct changes in specific peptides and proteins. While the changes in proteins may be very challenging to assess reproducibly, changes in peptides should be detectable, due to the ability to analyze the whole peptide without any manipulation/derivatisation, like tryptic digest or specific labelling.

Reviewer 4 Report
The paper, entitled " A novel urinary proteomics classifier for non-invasive evaluation of interstitial fibrosis and tubular atrophy in chronic kidney disease", highlights the importance of peptidome analysis. The paper is well written and has the merit of publication. However, there are some issues:
-Material and methods should be clarified: for analysis pooled and not individual? Number of peptides…
-A resume integrative picture could be done to provide an illustration of data and transpose it.
-Results could be better explored: e.g. Motif analysis, should be perform which proteases are involved?
Author Response
Review 4
The paper, entitled " A novel urinary proteomics classifier for non-invasive evaluation of interstitial fibrosis and tubular atrophy in chronic kidney disease", highlights the importance of peptidome analysis. The paper is well written and has the merit of publication. However, there are some issues:
- Material and methods should be clarified: for analysis pooled and not individual? Number of peptides
The analysed were performed individual for each sample, not pooled. Within the 435 analysed samples 4080 peptides with available sequence information were taken into consideration for statistical analysis. This information was added to the manuscript
- A resume integrative picture could be done to provide an illustration of data and transpose it
We thank the reviewer for this interesting idea. As we present very descriptive data and we try to discuss possible mechanistical links in the discussion. The vast range of peptides present within our classifier make it hard to think of one integrative picture to capture all aspects touched in this paper. Though, we agree that illustrating data would generate a more reader friendly environment.
- Results could be better explored: e.g. Motif analysis, should be perform which proteases are involved?
We would like to thank the Reviewer for this nice suggestion. We performed the analysis for prediction of proteases involved in the peptide generation. All predicted proteases are listed in table 3. We included the according methods, results and discussion into the manuscript.

Round 2
Reviewer 2 Report
Authors did not correct abstract as recommended
They should clearly state major finding in ABSTRACT conclusion
For example, they should state in the ABSTRACT conclusion that:
“ Fibrosis peptide profile FPP_29BH These 29 pep- 264 tides corresponded to a total of 13 different proteins. 19 of the 29 peptide fragments cor- 265 responded to 7 different collagen chains. 266 The classifier peptides and their properties are listed in table 2.
There are 29 peptide fragments of 13 different pro- 376 teins.
Alpha-2-HS-glycoprotein (AHSG), also known as fetuin-A, is a plasma binding pro- 400 tein. In our study, 4 different peptide fragments of fetuin-A were implemented in the fi- 401 brosis classifier containing 29 urinary peptides. “
Author Response
Dear reviewer,
We changed the Abstract results and conclusion to clearly state major findings. We tried to implement all your suggestions. We thank you for your helpful commentary.
Kind regards
Reviewer 3 Report
Dear editors and authors,
I endorse the reply.
Congratulations.
Author Response
Dear reviewer,
thank you for your helpful commentary
Kind regards
Reviewer 4 Report
author have accepted suggestions.
Author Response

(The authors gave the same response as above.)
